# Changes in Physical Activity and Sedentary Behavior in Response to COVID-19 and Their Associations with Mental Health in 3052 US Adults

**DOI:** 10.3390/ijerph17186469

**Published:** 2020-09-05

**Authors:** Jacob Meyer, Cillian McDowell, Jeni Lansing, Cassandra Brower, Lee Smith, Mark Tully, Matthew Herring

**Affiliations:** 1Department of Kinesiology, Iowa State University, Ames, IA 50011, USA; jenil@iastate.edu (J.L.); cbrower@iastate.edu (C.B.); 2The Irish Longitudinal Study on Aging and School of Medicine, Trinity College Dublin, The University of Dublin, Dublin D02 R590, Ireland; Cillian.McDowell@tcd.ie; 3Cambridge Centre for Sport and Exercise Sciences, Anglia Ruskin University, Cambridge CB1 2LZ, UK; Lee.Smith@anglia.ac.uk; 4Institute of Mental Health Sciences, Ulster University, Coleraine BT37 0QB, UK; m.tully@ulster.ac.uk; 5Physical Activity for Health Research Cluster, Health Research Institute, Department of Physical Education and Sport Sciences, University of Limerick, V94 T9PX Limerick, Ireland; Matthew.Herring@ul.ie

**Keywords:** COVID, physical activity, screen time, sitting time, sedentary, mental health, public health, depression, anxiety, loneliness

## Abstract

The COVID-19 pandemic altered many facets of life. We aimed to evaluate the impact of COVID-19-related public health guidelines on physical activity (PA), sedentary behavior, mental health, and their interrelations. Cross-sectional data were collected from 3052 US adults 3–8 April 2020 (from all 50 states). Participants self-reported pre- and post-COVID-19 levels of moderate and vigorous PA, sitting, and screen time. Currently-followed public health guidelines, stress, loneliness, positive mental health (PMH), social connectedness, and depressive and anxiety symptoms were self-reported. Participants were grouped by meeting US PA guidelines, reporting ≥8 h/day of sitting, or ≥8 h/day of screen time, pre- and post-COVID-19. Overall, 62% of participants were female, with age ranging from 18–24 (16.6% of sample) to 75+ (9.3%). Self-reported PA was lower post-COVID among participants reporting being previously active (mean change: −32.3% [95% CI: −36.3%, −28.1%]) but largely unchanged among previously inactive participants (+2.3% [−3.5%, +8.1%]). No longer meeting PA guidelines and increased screen time were associated with worse depression, loneliness, stress, and PMH (*p* < 0.001). Self-isolation/quarantine was associated with higher depressive and anxiety symptoms compared to social distancing (*p* < 0.001). Maintaining and enhancing physical activity participation and limiting screen time increases during abrupt societal changes may mitigate the mental health consequences.

## 1. Introduction

The novel coronavirus (COVID-19) has rapidly altered many facets of life globally. In the US, all 50 states and the federal government had made emergency declarations by 16 March 2020. In response to this global pandemic, governments have introduced diverse measures [1] designed to limit disease transmission to prevent critically overburdening healthcare systems. These measures range from social or physical distancing (staying ≥6 feet/2 m away from others) to quarantining people who have been exposed to the virus for 14 days or longer. Changes in work and social environments occurred rapidly and likely influenced both behavior and mental health, but limited data exist to determine the impact of these changes.

The effects of making pandemic-related behavioral changes on population mental health are not well documented. A 2020 rapid review [2] found that quarantine regularly resulted in acute negative psychological effects with potentially persistent effects. Recent cross-sectional surveys from adults in China indicated high levels of depressive and anxiety symptoms likely associated with the pandemic [3,4,5]. Furthermore, physically active people reported being more impacted psychologically by COVID-19 response measures in China [6], potentially due to limited opportunities for activity.

Physical activity appears to be reduced following COVID-related public health restrictions. A recent blog post from Fitbit Inc. indicated average decreases in step count across the US during the week of 22 March of 12%, with larger decreases across the world [7] which is mirrored in recent data from Azumio [8]. As only 26 percent of men and 19 percent of women report meeting the US physical activity guidelines [9], and there are consistent positive benefits of regular physical activity for mental health [10,11,12], reductions in physical activity are likely to compound the already-problematic psychological effects of the pandemic. As 19.1 percent of US adults were estimated to have a mental illness in the past year [13], psychological health is already a major concern in the US. Finding and promoting ways to improve or maintain psychological health have been encouraged [14]. Being regularly physically active could limit the impact of the pandemic on mental health. However, data are not yet available to indicate the associations between changes in physical activity and sedentary behavior due to pandemic-related public health restrictions and mental health.

Given the rapidly evolving response to COVID-19 and the paucity of current data, the present study was designed and conducted to evaluate the impact of COVID-19-related public health guidelines on physical activity (PA), sedentary behavior, mental health, and their interrelations. Specifically, we aimed to evaluate three hypotheses: (1) that self-reported changes in physical activity, sitting time, and screen time after the pandemic would occur relative to the degree of COVID-related public health restrictions that were followed, (2) that self-reported current mental health would be associated with the degree of changes in physical activity, sitting time, and screen time (a) and COVID-related public health restrictions (b), and, (3) that the association between changes in physical activity and current mental health would be moderated by the degree of COVID-related public health restrictions that were followed. Evaluating these hypotheses will critically inform current and future policy approaches related to pandemics.

## 2. Materials and Methods

The design of the ‘COVID-19 and Wellbeing’ study includes cross-sectional and longitudinal components which were approved as an exempt project by the Iowa State University Institutional Review Board (IRB# 20-144-00) and is associated with a broader cross-national collaborative effort focused on self-isolation. Cross-sectional data were investigated herein. Convenience sampling using mass emails that included a link to an anonymous online survey to Iowa State University students, faculty, staff, and alumni, snowball sampling (i.e., participants recruiting others), and posts to social media pages were used to recruit self-selected participants (Figure 1). Data analyzed were collected 3–7 April 2020. This study adhered to Strengthening the Reporting of Observational Studies in Epidemiology (STROBE) guidelines [15].

Inclusion criteria were age of ≥18 years and current US residence. Potential participants provided informed consent and confirmed inclusion criteria before starting the survey. Participants self- reported demographic information, health history, COVID-19-related restrictions they were following, COVID-19-related health behaviors and their changes, and mental health questionnaires.

### 2.1. Demographics and Health History

Participants self-reported age, gender, sex, race, education, marital status, occupational status, height and weight. BMI was calculated from self-reported height and weight. Health history included self-reported current chronic health conditions based on a list of common illnesses.

### 2.2. COVID-19-Related Public Health Restrictions

As different localities provided different guidance on what types of behaviors were required to be followed and what were recommended to be followed (and this study included respondents throughout the US), participants were allowed to self-select which public health restrictions they were currently following. Participants were provided with the following information to determine their individual circumstances:

The following questions ask specifically about what preventative and mitigating measures you are implementing with COVID-19. For these questions, use the following definitions:
Self-Isolation: For people who actually have the virus or suspect they may be infected. People who have been infected with the virus may be asked to self-isolate at home if they have no symptoms or are only mildly ill.Quarantine: For those who may have been exposed to the virus. They are asked to stay at home. Some people may choose to be asked to self-quarantine, meaning they do it voluntarily because they think they may have been exposed or they are being cautious.Shelter-in-place: People that are being asked to stay at home as much as possible, meaning they shouldn’t be out unless getting food, gas, or other essentials, or for medical reasons.Stay-at-home order: Residents can still go out for essential needs as long as they are practicing social distancing and “common sense”.Social distancing: means remaining out of congregate settings, avoiding mass gatherings, and maintaining distance (approximately 6 feet or 2 m) from others when possible.


Collectively, changes to your behavior that have been made related to any of these will be called “COVID-related behavioral changes”.

Participants indicated which public health restrictions they were currently following by selecting all that applied: quarantined, self-isolating, under a shelter-in-place, stay-at-home order, and social distancing. Participants were grouped based upon the most significant restriction that they were following, grouping quarantined and self-isolation as the most restrictive, shelter-in-place or stay-at-home next, and social distancing as the least restrictive.

### 2.3. COVID-19-Related Health Behaviors and Change

Participants reported current smoking status. Participants reported average daily time spent sitting, engaged in moderate and vigorous physical activity (reported separately), and average daily screen-time. These were reported based on asking about these behaviors both pre- and post- restrictions.

### 2.4. Mental Health

The 4-item Perceived Stress Scale-4, (range: 0–16) assessed stress; higher scores indicate greater perceived levels of stress (α = 0.60–0.82) [16].

The 3-item Loneliness scale examined loneliness (range 0–3); higher scores indicate greater loneliness. This measure has demonstrated high internal consistency in previous studies (α = 0.72) [17].

The Short Warwick–Edinburgh Mental Wellbeing Scale (SWEMWBS-7; range 7–35) examined positive mental health (PMH); higher scores indicate more positive mental health. This scale has demonstrated high internal consistency in other populations (Cronbach’s α = 0.83–0.87) [18].

Social engagement was assessed using a 3-item form of the Lubben Social Network Scale-6 that combined friends and relatives in individual questions (range 0–15); higher scores indicate greater social engagement [19].

The psychometrically strong (α = 0.91) [20] 21-item Beck Depression Inventory-II (BDI) [21], excluding the suicidality question (20 items total), assessed depressive symptoms. Total scores were divided by 20, then multiplied by 21. Individuals were classified: minimal depressive symptoms (0–13), mild depressive symptoms (14–19), moderate depressive symptoms (20–29), or severe depressive symptoms (30–63).

The psychometrically strong (α = 0.92, r = 0.75) 21-item Beck Anxiety Inventory (BAI) assessed anxiety symptoms [22]. Scores range from 0 to 63. Individuals were classified: low anxiety (0–21), moderate anxiety (22–35), or potential concerning anxiety levels (36–63).

### 2.5. Statistical Analysis

Analyses were performed using Stata (v14.2; StataCorp., College Station, TX, USA). Participant characteristics were described by means and standard deviations (SDs) for continuous variables and proportions for categorical variables. Participants were categorized according to meeting US Physical Activity Guidelines [9], reporting ≥8 h/day of sitting, or reporting ≥8 h/day of screen time (as in [23]) both pre-/post-COVID-19 public health restrictions. Participants were then classified as “maintaining low physical activity” if they did not adhere to the guidelines at either timepoint, as “increasing physical activity” if they did not adhere to the guidelines prior to restrictions but did afterwards, etc. Participants were similarly classified for sitting and screen time.

To test Hypothesis 1, differences in physical activity, sitting time, and screen time pre- /post- COVID-19 public health restrictions, stratified by physical activity status prior to the restrictions, were quantified by Hedges’ *g* effect sizes and associated 95% confidence intervals (95% CIs), and calculated with increased time in each behavior represented as a positive effect size [24]. These were converted to percentages of pre-COVID-19 behavior times for ease of interpretation in Figure 2. Differences were categorized as “clinically meaningfully” when *g* was ≥ 0.50 [25]. To test Hypotheses 2a and 2b, multivariable linear regression quantified associations (adjusted unstandardized betas (b) and associated SEs) of groups based on change in physical activity, sitting time, and screen time, and public health restrictions, with continuous depressive symptoms, anxiety symptoms, loneliness, stress, social network, and PMH. To test hypothesis 3, multivariable linear regressions were re-run including interaction terms (physical activity change X public health restrictions, sitting time change X public health restrictions, and screen time change X public health restrictions). All linear regressions included age, sex, race, BMI (continuous), smoking status, marital status, employment status, and presence of chronic disease.

Multicollinearity was determined as likely if two covariates had a correlation ≥0.8, the mean variance inflation factor was ≥6, or the highest individual variance inflation factor was ≥10. The variance inflation factors between the maintained low and maintained high categories for sitting time, screen time, and physical activity were all below 1.87 indicating minimal multicollinearity. For the present study, the highest correlation between two covariates was 0.52, the mean variance inflation factor was 2.56, and the highest individual variance inflation factor was for education at 15.7. Consequently, education was excluded from the linear regressions. Robust standard errors, which are robust to heteroscedasticity, were also used in the multivariable linear regressions. To adjust for multiple testing (Hypotheses 2a and 3: three independent variables and six dependent variables; Hypothesis 2b: one independent variable and six dependent variables), statistical significance was established as *p* < 0.00833 for Hypotheses 2a and 3 and *p* < 0.00278 for Hypothesis 2b.

## 3. Results

### 3.1. Participant Characteristics

As of 9:30 am Central Daylight Time on 8 April, a total of 4542 entries had been started, with 3242 participants consenting and completing the project and, after excluding those missing exposure or outcome data, a total of 3052 with complete data were analyzed (Figure 1) for a completion rate of 71.4% with 67.9% after exclusions. Participant characteristics are presented in Table 1. Briefly, participants (*n* = 3052; 62% female) were relatively evenly dispersed from ages 18–75+, predominantly white and educated, and overweight but mostly without any chronic conditions. Mean ± SD outcome scores in the total population were: depressive symptoms (9.44 ± 8.49), anxiety symptoms (7.29 ± 8.08), loneliness (5.12 ± 1.81), stress (6.07 ± 3.00), social network (8.52 ± 2.64), and PMH (24.30 ± 4.65).

### 3.2. Change in Physical Activity, Sitting Time, and Screen Time

Mean percentage change in physical activity, sitting time, and screen time among participants who met and did not meet minimum recommended levels of physical activity prior to COVID-19 restrictions are presented in Figure 2, and stratified by the levels of restrictions they are experiencing.

Among active participants pre-COVID-19 restrictions, those in social isolation showed the largest (and clinically meaningful) drop in physical activity (*g* = −0.913 [95%CI: −1.088 to −0.739]) and increase in sitting (*g* = 0.698 [0.526 to 0.869]) and screen time (*g* = 0.653 [0.482 to 0.823]). Among those with stay-at-home and social distancing restrictions, changes in physical activity (stay-at-home: *g* = −0.555 [−0.667 to −0.443]; social distancing: *g* = −0.514 [−0.647 to −0.381]), sitting time (stay-at-home: *g* = 0.485 [0.374 to 0.597]; social distancing: *g* = 0.511 [0.378 to 0.643]), and screen time (stay-at-home: *g* = 0.529 [0.417 to 0.640]; social distancing: *g* = 0.559 [0.426 to 0.692]) were comparable.

Among inactive participants pre-COVID-19 restrictions, those in social isolation also self- reported the largest and clinically meaningful increases in sitting (*g* = 0.565 [0.393 to 0.735]) and screen time (*g* = 0.589 [0.417 to 0.760]). Among those with stay-at-home and social distancing restrictions, changes in sitting (stay-at-home: *g* = 0.391 [0.294 to 0.488]; social distancing: *g* = 0.311 [0.196 to 0.426]) and screen time (stay-at-home: *g* = 0.437 [0.340 to 0.535]; social distancing: *g* = 0.421 [0.306 to 0.536]) were comparable. In contrast to the active participants, no change in physical activity was observed in the previously inactive participants (self-isolation: *g* = −0.101 [−0.269 to 0.067]; stay- at-home: *g* = 0.071 [−0.026 to 0.167]; social distancing: *g* = 0.022 [−0.092 to 0.135]).

### 3.3. Associations between Changes in Behavior, COVID-19 Public Health Restrictions, and Mental Health

Associations between changes in physical activity, sitting time, and screen time pre-/post- COVID-19 related public health restrictions and mental health outcomes in the total population are presented in Table 2. Statistically significant results are outlined here. Compared to those who maintained adherence to the physical activity guidelines, those who decreased (i.e., moved from active to inactive) had stronger/higher depressive symptoms (adjusted unstandardized beta: b = 1.960; *p* < 0.001), loneliness (b = 0.340; *p* < 0.001), and stress (b = 0.522; *p* < 0.001), and lower PMH (b = −1.010; *p* < 0.001). Those who maintained low physical activity levels had lower levels of social network (b = −0.389; *p* = 0.001) and PMH (b = −0.629; *p* < 0.001) and higher levels of stress (b = 0.377; *p* = 0.002).

Results were similar for screen time. Compared to those who maintained screen time <8 h/day (i.e., maintained “low” screen time), those who increased had higher depressive symptoms (b = 1.924; *p* < 0.001), loneliness (b = 0.340; *p* < 0.001), and stress (b = 0.590; *p* < 0.001), and lower PMH (b = −0.920; *p* < 0.001). Sitting time was not significantly associated with any outcome.

Compared to those social distancing, those in self-isolation had higher depressive (b = 1.427; *p* < 0.001) and anxiety symptoms (b = 1.640; *p* < 0.001; Appendix A). Full model results are in Appendix A. Analyses for the third hypothesis showed that public health restrictions did not moderate associations between activity behaviors and mental health (all *p* ≥ 0.003; Appendix A).

## 4. Discussion

This manuscript presents a timely investigation of changes in physical activity, sitting time, and screen time as a result of COVID-19 public health restrictions, and their associations with mental health. The current findings indicate: (1) large reductions of physical activity and increases in sedentary time across the population and particularly among previously physically active and self-isolated/quarantined individuals; (2) consistent associations between reductions in physical activity and increases in screen time with higher negative mental health and lower positive mental health; and, (3) more severe anxiety and depressive symptoms for those in self-isolation compared to less restrictive situations, which were not moderated by changes in physical activity or sedentary behavior. Some models suggest persistent physical distancing may be required for three months, and possibly for eighteen months, to mitigate the peak effects of COVID-19 on health systems [26]. Recent data also indicate that previous physical inactivity (assessed in 2006–2010) was associated with a 32% increased risk of hospitalization from COVID-19 in the UK Biobank study, highlighting the potential importance of maintaining or increasing physical activity [27]. Together, these findings strongly support the need to facilitate and promote physical activity and limit increases in screen time throughout the duration of pandemic-related or other major public health-related restrictions, however long they may be required.

Participants who met the physical activity guidelines prior to COVID-19-related restrictions decreased their physical activity by 32%, on average, with those in self-isolation reported the greatest decrease of 43%. The magnitude of changes in physical activity and screen time found here are potentially meaningful based on a commonly utilized important difference of 0.5 standard deviation unit [25]. Unsurprisingly, no significant change in physical activity was seen among those who were not active prior to COVID-19-related restrictions. This extends data released by Fitbit and from that collected through Azumio that show substantial decreases in objectively-monitored physical activity in the US and across the world [7,8]. However, previous data were not stratified based on prior physical activity levels. Concerningly, previous research has shown that preventing people from exercise was consistently associated with increases in depressive and anxiety symptoms, with larger increases seen when withdrawal lasted more than two weeks [28]. Thus, maintaining or increasing physical activity during periods of significant societal changes could have profound effects on sustaining mental health.

Physical activity has well-established inverse associations with anxiety and depressive symptoms [10,11,12,29], and recent evidence showed inverse associations between physical activity and depressive symptoms among Vietnamese adults with suspected COVID symptoms [30]. However, dynamic associations between physical activity and mental health over short time periods are less studied. Previous prospective cohort studies demonstrated physical activity and mental health associations over prolonged periods of time; however, such rapid, large, potentially clinically meaningful changes to physical activity as shown herein and on a population scale is unprecedented, and the health effects are relatively unknown. Previously, experimentally decreasing physical activity among active adults can have significant impacts on depression and mood after just one week [31]. Consistent with these previous findings, participants who self-reported being previously active who no longer reported being active following COVID-19-related public health restrictions reported worse mental health across almost all evaluated dimensions compared to those who maintained their activity level. The present findings support concerted efforts to promote opportunities for regular physical activity to preserve mental health among previously physically active adults and potentially enhance mental health among both physically active and inactive adults. Potential approaches could include telehealth interventions or public broadcasting time devoted to promotion/implementation of home-based physical activity to facilitate activity among vulnerable populations and those in isolation.

The lack of behavior by public health restriction interactions on mental health was not unexpected. This indicates that the way that people’s behavior changed did not significantly alter the association of public health restrictions with mental health. It is possible that self-isolation/quarantine was associated with consistently lower mental health regardless of behavior changes, or that the effects of self-isolation/quarantine on sitting, screen or active time were consistent enough across people so that potential interactions were not found. Overall, the associations between mental health and changing physical activity and screen time underline the importance of these behaviors regardless of the specific public health restrictions that are in place.

Much past research has conceptualized mental health based on the presence/absence of negative symptoms (e.g., depressive and anxiety symptoms); the positive mental health benefits of physical activity are currently understudied. A recent study of 5090 Finnish adults reported that physical inactivity overall (and particularly leisure-time physical inactivity) and long screen time at home, were associated with higher odds of low positive mental health [32]. The present results expand past associations by indicating that people who reported screen time increases, or whose physical activity decreased or remained low, had lower positive mental health. As 68.9% of the present sample reported either decreasing activity or maintaining low activity, the lower positive mental health in these groups is of public health concern.

Similarly, substantial increases in sitting and screen time were observed. Evidence regarding the mental health impacts of sitting and screen time is mixed, and the effects of such large, acute increases in sedentary behaviors are unknown. Presently, participants who increased their screen time reported higher negative mental health and lower positive mental health across almost all evaluated dimensions compared to those who maintained lower levels. However, no associations between sitting time and mental health were observed. It is plausible that the differing mental health effects of mentally-active and mentally-passive sedentary behaviors explain this distinction. In a cohort of 24,000 Swedish adults, substituting mentally-active sedentary behavior for mentally-passive behavior was associated with a reduced risk of developing major depression over thirteen years [33]. Screen time is commonly defined as a mentally-passive sedentary behavior, potentially explaining the consistent observed associations between screen time and mental health. The large and rapid changes in screen time reported herein (over weeks rather than months or years) indicate acute health-related effects of increased screen time. There are likely required increases in screen time due to shifts from in-person to remote, screen-based work to adhere to COVID-19 restrictions. With a large transition to virtual work- or school-environments, limiting non-work/school screen time and balancing increased screen usage with opportunities to be active will be paramount for maintaining mental health.

### Strengths and Limitations

These findings should be considered in the context of their strengths and limitations. Strengths include data on physical activity and sedentary behavior pre- and post-COVID-19 public health restrictions, evaluation of both physical activity and sedentary behavior, and the use of well- validated measures of mental health in a large sample of US males and females across broad age demographics. Nonetheless, the cross-sectional design precludes inference of causality and the sample is predominantly well-educated and white. In particular, as this study sample is not representative of the entire US population, generalizations should be limited based on the sample characteristics. All behaviors and currently followed public health guidelines were self-reported and included a recall of pre-COVID-19 activity, which is potentially subject to misreporting. The self- selection and convenience sampling of participants to complete the survey may also affect the results, although a >70% completion rate for those who began the survey is high. Further, while it was expected that certain demographic factors (e.g., female, age, chronic conditions) were associated with mental health, the influence of changing employment status should be further explored. Finally, how these behaviors change across time and the prospective relationships between health behaviors and mental health are of public health interest, but the current cross-sectional and retrospective design preclude their evaluation.

## 5. Conclusions

The current findings strongly support the mental health benefits of implementing measures that promote physical activity while limiting screen time during periods of societal modification due to a pandemic. Potentially effective methods to do so may be through enhanced telehealth or public broadcasting time devoted to promotion/implementation of home-based physical activity. Future research should replicate these findings in other large samples, investigate potential cross-national differences, longitudinally assess dynamic relationships between these factors, and integrate device- based measures.

## Figures and Tables

**Figure 1 ijerph-17-06469-f001:**
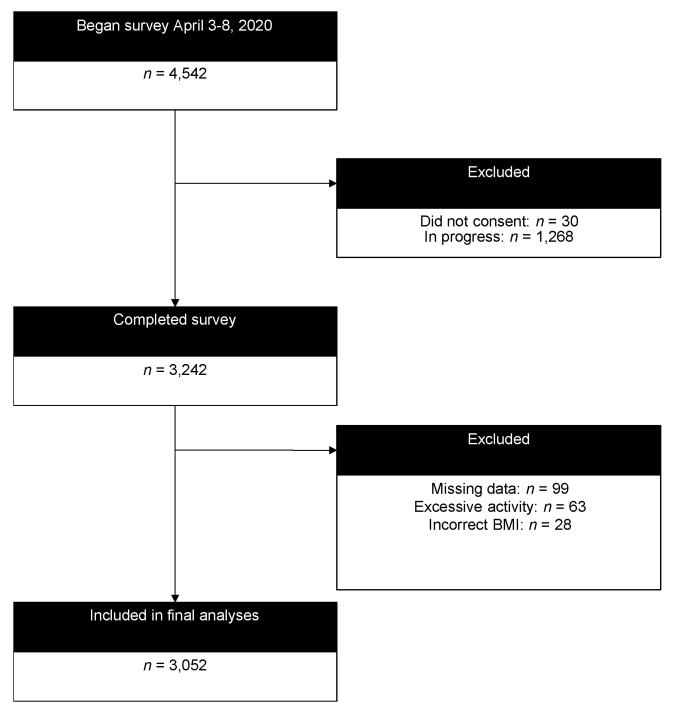
CONSORT diagram of participation.

**Figure 2 ijerph-17-06469-f002:**
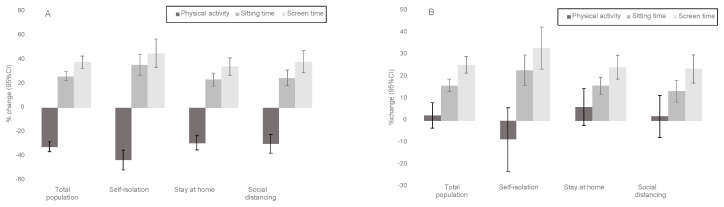
Mean percentage change (95% CI) in behaviors from before to after COVID-19 related public health restrictions in those who were previously (**A**) active and (**B**) inactive. Panel A shows those who met the minimum recommended physical activity levels prior to the restrictions (*n* = 1361) by public health restriction category (i.e., self-isolation: *n* = 278; stay at home: *n* = 635; social distancing: *n* = 448), while Panel B shows those who did not meet the minimum recommended physical activity levels prior to the restrictions (*n* = 1691) by public health restriction category (i.e., self-isolation: *n* = 272; stay at home: *n* = 827; social distancing: *n* = 592.

**Table 1 ijerph-17-06469-t001:** Participant characteristics (*n* = 3052).

Characteristic	*N* (%) or Mean ± SD
Age	
18–24	508 (16.64)
25–34	470 (15.40)
34–44	419 (13.73)
45–54	376 (12.32)
55–64	474 (15.53)
65–74	522 (17.10)
75+	283 (9.27)
Sex	
Male	1151 (37.63)
Female	1897 (62.01)
Transgender	4 (0.13)
Race (white)	2848 (93.10)
BMI	26.84 ± 5.64
Underweight (BMI < 18.5)	53 (1.74)
Normal (BMI 18.5–25)	1281 (41.97)
Overweight (BMI 25–30)	990 (32.44)
Obese (BMI > 30)	728 (23.85)
Smoker	80 (2.62)
Marital Status	
Married/in a relationship	2070 (67.67)
Widowed	93 (3.04)
Separated/divorced	178 (5.82)
Never married	711 (23.24)
Education	
Up to high school graduate	56 (1.83)
Up to college graduate	1656 (54.14)
Graduate degree	1340 (43.81)
Employment	
Employed	1747 (57.11)
Retired	785 (25.66)
Unemployed	403 (13.17)
Other	97 (3.17)
Chronic Conditions	
0	2163 (70.71)
1	263 (8.60)
2+	626 (20.46)
Depression	
Minimal	2368 (77.59)
Mild	375 (12.29)
Moderate	217 (7.11)
Severe	92 (3.01)
Anxiety	
Low	2836 (92.92)
Moderate	183 (6.00)
High	33 (1.08)

BMI = body mass index; SD = standard deviation.

**Table 2 ijerph-17-06469-t002:** Adjusted associations between self-reported changes in behavior from pre- to post-COVID-19-related restrictions and current mental health.

		Depression		Anxiety		Loneliness		Stress		Social Network		Positive Mental Health	
		Adjusted R^2^	*p*-Value	Adjusted R^2^	*p*-Value	Adjusted R^2^	*p*-Value	Adjusted R^2^	*p*-Value	Adjusted R^2^	*p*-Value	Adjusted R^2^	*p*-Value
**Goodness of Fit**		0.268	<0.0001	0.219	<0.0001	0.168	<0.0001	0.202	<0.0001	0.046	<0.0001	0.255	<0.0001
	***n***	**b (SE)**	***p*-Value**	**b (SE)**	***p*-Value**	**b (SE)**	***p*-Value**	**b (SE)**	***p*-Value**	**b (SE)**	***p*-Value**	**b (SE)**	***p*-Value**
**Physical Activity**													
Maintained high	798	REF		REF		REF		REF		REF		REF	
Increased	152	−0.505 (0.645)	0.434	0.066 (0.677)	0.923	0.001 (0.136)	0.996	−0.133 (0.244)	0.585	0.305 (0.222)	0.169	0.189 (0.345)	0.585
Decreased	563	**1.960 (0.417)**	**<0.001**	0.596 (0.411)	0.148	**0.340 (0.096)**	**<0.001**	**0.522 (0.155)**	**<0.001**	−0.269 (0.149)	0.072	**−1.010 (0.230)**	**<0.001**
Maintained low	1539	0.629 (0.318)	0.048	0.248 (0.320)	0.439	0.078 (0.075)	0.302	**0.377 (0.124)**	**0.002**	**−0.389 (0.119)**	**0.001**	**−0.629 (0.182)**	**<0.001**
**Sitting Time**													
Maintained low	1041	REF		REF		REF		REF		REF		REF	
Decreased	85	0.587 (0.972)	0.546	1.497 (0.944)	0.113	−0.067 (0.197)	0.728	−0.015 (0.329)	0.964	0.074 (0.273)	0.786	−0.122 (0.505)	0.809
Increased	582	0.918 (0.441)	0.037	0.946 (0.445)	0.034	0.195 (0.097)	0.045	0.253 (0.154)	0.102	−0.163 (0.149)	0.275	−0.673 (0.234)	0.005
Maintained high	1344	−0.199 (0.348)	0.566	−0.064 (0.344)	0.852	0.046 (0.078)	0.554	−0.068 (0.131)	0.604	−0.040 (0.126)	0.750	−0.036 (0.192)	0.853
**Screen Time**													
Maintained low	1512	REF		REF		REF		REF		REF		REF	
Decreased	45	−0.623 (1.095)	0.569	0.345 (1.203)	0.774	−0.495 (0.198)	0.013	−0.261 (0.468)	0.577	0.762 (0.393)	0.052	0.412 (0.641)	0.520
Increased	562	**1.924 (0.441)**	**<0.001**	1.341 (0.454)	0.003	**0.340 (0.095)**	**<0.001**	**0.590 (0.154)**	**<0.001**	−0.069 (0.145)	0.632	**−0.920 (0.239)**	**<0.001**
Maintained high	933	0.375 (0.392)	0.339	0.474 (0.375)	0.206	0.146 (0.085)	0.087	0.126 (0.137)	0.361	−0.156 (0.133)	0.243	−0.451 (0.202)	0.026

Physical activity, sitting time, and screen time were entered in the model simultaneously and adjusted for age, sex, race, smoking, relationship status, employment, chronic illnesses, and COVID-19 public health restrictions. Education was excluded due to multicollinearity. b = adjusted unstandardized beta; SE = standard error. Black text and bold-face indicates statistical significance, set at *p* < 0.00278.

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
