# Peer review of "Changes in Physical Activity and Sedentary Behavior in Response to COVID-19 and Their Associations with Mental Health in 3052 US Adults"

_ijerph, 2020, doi:10.3390/ijerph17186469_

Round 1
Reviewer 1 Report
This study aims to evaluate the impact of COVID-19-related public health guidelines on physical activity, sedentary behavior, mental health, and their interrelations.
The reference style does not follow the guidelines of the paper
I strongly recommend the authors seek English language revision for this manuscript. I believe this would help clarify some of the expressions and sentences that are currently not appropriate or incomprehensible.
Some references are old: more than 15 years
Abstract:
“Participants self-reported moderate and vigorous PA, sitting, and screen time, both pre- and post-COVID-19-related restrictions.” ??? The sentence not complete
Introduction: add some information about the prevalence of physical activity and mental health in the US
Methods:
- Ethical approval ??
Reviewer 2 Report
This interesting article aimed to evaluate the impact of 18 COVID-19-related public health guidelines on physical activity (PA), sedentary behavior, mental 19 health, and their interactions with each other.
I would like to congratulate the authors for the quality of their work and the great information provided on such a current topic as COVID.19 However, I am enclosing some slight changes that may improve the article:
1. INTRODUCTION
- Authors should include the aim of the study (although it already appears in the abstract).
2. METHODS
- Was the sampling selected by 'snowball'?
- How did the authors calculate the height, weight, BMI? It was self-reported? All information on anthropometric data must be included.
- COVID-19-related public health restrictions. This section should be detailed, as each country has adopted different measures regarding movement restrictions.
3. RESULTS
- The number of overweight and obese people should be included.
- I would like the authors to indicate whether the changes in the BMI have been included as a covariate.
4. DISCUSSION
- The limitations should be extended. There are many important aspects that have not been mentioned.
Congratulations again to the authors for their brilliant work.
Best wishes,
Reviewer 3 Report
Thank you for send this review to me. I enjoyed reading it.
There are a few minor issues that need to be clarified.
Lines 92-94 Are quarantined and self-isolated the same. It would be less confusing if the were written with a / between them rather than the word "or". I preseume stay at home is the same as "in a place of shelter". If so the same style shuold apply.
Line 144- was collinearity tested between physical activity, sitting time and screen time? If so it would be helpful to say this because intuitively I thought there might be collinearity.
Lines 185-188 and 189-195 are very difficult to decipher as they seem jumbled. It might be easiest to just list those areas in which there were significant before and after changes eg for those self-isolating there were significant reductings in a and b, for those staying at home etc....
Lines 2224-226 seem redundant or if the point is rleevant it should be expressed why that is the case.
Lines 226-231 says that reduced activity is associated with a higher risk of admission due to Covid. Surely this is likely to be due to the effect of the infection on energy rather than the toerh way round. this makes the suggestion that improving exercise will reduuce admission risk redundant or perhaps I am misinterpreting the sentence.
The authors might consider wish to discuss why there were no interactions - was this a surprise?
Round 2
Reviewer 1 Report
Thank you for the revision!
Reviewer 2 Report
The authors have made all the modifications indicated. Without a doubt, the article deals with a very interesting topic on a very topical and interesting subject. To my mind, the article is ready to be published.
Best wishes,